# Complete Genome Expression Analysis of the Auxin Response Factor Gene Family in Sandalwood and Their Potential Roles in Drought Stress

Xiaojing Liu [1,2,†], Yunshan Liu [1,2,†], Shengkun Wang [1], Fangcuo Qin [1], Dongli Wang [1], Yu Chen [2], Lipan Hu [2], Sen Meng [1] and Junkun Lu [1,*]

1   State Key Laboratory of Tree Genetics and Breeding, Research Institute of Tropical Forestry, Chinese Academy of Forestry, Guangzhou 510520, China
2   College of Biology and Food Engineering, Chongqing Three Gorges University, Wanzhou, Chongqing 404100, China
*   Correspondence: junkunlu@caf.ac.cn
†   These authors contributed equally to this work.

**Abstract:** Auxin response factors (ARFs) are essential transcription factors in plants that play an irreplaceable role in controlling the expression of auxin response genes and participating in plant growth and stress. The *ARF* gene family has been found in *Arabidopsis thaliana*, apple (*Malus domestica*), poplar (*Populus trichocarpa*) and other plants with known whole genomes. However, *S. album* (*Santalum album* L.), has not been studied. In this study, we analyzed and screened the whole genome of *S. album* and obtained 18 *S. album ARFs* (*SaARFs*), which were distributed on eight chromosomes. Through the prediction of conserved domains, we found that 13 of the 18 SaARFs had three intact conserved domains, named DBD, MR, Phox and Bem1 (PB1), while the extra five SaARFs (SaARF3, SaARF10, SaARF12, SaARF15, SaARF17) had only two conserved domains, and the C-terminal PB1 domain was missing. By establishing a phylogenetic tree, 62 *ARF* genes in *S. album*, poplar and *Arabidopsis* were divided into four subgroups, named I, II, III and IV. According to the results of collinearity analysis, we found that ten of the eighteen *ARF* genes were involved in five segmental duplication events and these genes had short distance intervals and high homology in the *SaARF* gene family. Finally, tissue-specific and drought-treatment expression of *SaARF* genes was observed by quantitative real-time polymerase chain reaction (qRT-PCR), and six genes were significantly overexpressed in haustorium. Meanwhile we found *SaARF5*, *SaARF10*, and *SaARF16* were significantly overexpressed under drought stress. These results provide a basis for further analysis of the related functions of the *S. album ARF* gene and its relationship with haustorium formation.

**Keywords:** *Santalum album* L.; auxin response factors; auxin; qRT-PCR; drought

## 1. Introduction

*Santalum album* L., a semiparasitic species of the genus Sandalwood in the family Sandalwood, is distributed mainly in India, Indonesia, and Australia [1,2]. In contrast to holoparasitic plants, hemiparasitic plants not only have the ability to perform photosynthesis, but also can grow autonomously, in some cases where no host plants exist [3]. Sandalwood is a rare tree species; its oil and hardwood have significant commercial value, used in perfumes, cosmetics, medicine, and aromatherapy, and recently in the prevention of skin cancer, but this has also led to a sharp decline in wild sandalwood populations [4–6]. In nature, there are more than 300 plants that can serve as sandalwood hosts and parasitic angiosperms depend on host root-derived chemical signals to control various stages of development [7,8]. The haustorium is a unique organ of parasitic plants; these parasite-specific structures penetrate host roots and connect the host and parasite xylem vessels [9]. Futhermore, their main function is to absorb water and nutrients from the host plant, especially during early phases



of development [3]. However, we also found that sandalwood sometimes grow haustoria even in the absence of a host plant.

Auxin, also known as indole-3-acetic acid, is widely used as the earliest hormone found to promote plant growth and can promote the growth of plant stems, shoots and roots [10–13]. At present, several genes related to auxin transduction have been found, including *Aux/IAA* (auxin/indole-3-acetic acid), *GH3* (gretchen hagen 3), *SAUR* (small auxin-up RNA), and *ARF* (auxin response factor) [12]. *ARF*, a transcription factor that regulates the expression of auxin response genes under the influence of auxin concentricon, can combine with AuxREs, the auxin response element in the promoter of target genes, to promote or inhibit the expression of auxin response genes [14]. ARF can also interact with Aux/IAA proteins, which also have auxin signal transduction functions. Auxin sensing begins with auxin binding to the TIR1 (TRANSPORT INHIBITOR RESPONSE1)/AFB (AUXIN SIGNALING F-box) receptor and leads to the subsequent degradation of Aux/IAA proteins that inhibit auxin signaling through physical interaction with ARF proteins by the ubiquitin family [15–17].

The ARF protein has three stable conserved domains, namely, the DBD, MR and Phox and Bem1 (PB1) domains [18]. The function of the DBD located at the N-terminus is to bind to AuxREs (TGTCTC cis-responsive elements), the promoters of auxin-responsive genes, to control the expression of target genes [19]. The middle MR domain is divided into an AD activation domain rich in glutamine (AtARF5, AtARF6, AtARF7, AtARF8, AtARF19) and an RD inhibition domain rich in serine and threonine (AtARF1-4, AtARF9-18, AtARF20-23) [20]. The ARF-MRs, which belong to the RD inhibitory domain, are divided into class B and C. Some MRs in class B with inhibitory domains have been found to also contain an amphiphilic inhibitory motif related to the ethylene response element binding factor the EAR motif [21], which can interact with the N-terminal part of the Aux/IAA protein. The MR domain is mainly responsible for controlling the function of ARFs. The PB1 domain located at the C-terminal is homologous to the C-terminal domain of the Aux/IAA protein, which can combine to form homologs or heterodimers [22], and its main function is to mediate protein-protein interactions, for example, between an ARF protein and Aux/IAA protein or between an ARF protein and another ARF protein. Auxin affects the activity of the ARF protein by controlling the Aux/IAA protein and then controls the expression of auxin response genes. At low auxin concentrations, the C-terminal domain of the Aux/IAA protein and the C-terminal domain of ARF protein were inhibited, and the N-terminal EAR domain of the Aux/IAA protein recruited the corepressor TOPLESS (TPL) [23], which affected the activity of the ARF protein and reduced the expression of auxin responsive genes. At high auxin concentrations, the Aux/IAA protein and a small amount of auxin are promoted to bind to the SCF$^{TIR1/AFB}$ corepressor complex, which is then decomposed by the ubiquitin family, thereby releasing more ARF protein to bind to auxin responsive genes and promote their expression [24].

As a large gene family, the *ARF* gene has been successively studied in *Arabidopsis*, maize, tomato, poplar and other plants. *Arabidopsis* contains 23 *ARF* genes [25], while rice (*Oryza sativa*), apple (*Malus*), poplar (*Populus trichocarpa*) and tomato (*Lycopersicon esculentum*) contain 25, 31, 39 and 21 *ARF* genes [26–29], respectively. However, the biological function of the *SaARF* gene family in response to auxin in sandalwood remains unclear. The related functions of *ARF* genes are generally analyzed in model plants such as *Arabidopsis thaliana* and rice [30]. For example, *AtARF2* is related to the synthesis of flavonoids and anthocyanins [31], and *AtARF7* and *AtARF19* are specifically distributed in roots and related to lateral root elongation [32]. *SlARF9* is one of the auxin-related genes differentially expressed in tomato fruit set and early fruit development [33]. The Aux/IAA9-ARF5 module regulates wood formation by coordinating the expression of HD-ZIP III transcription factors in poplar [34]. The miR167-*GmARF8* module plays a key role in the auxin-mediated nodule and lateral root formation in soybean [35]. The protein encoded by *ARF8* affects hypocotyl elongation and root behavior in *Arabidopsis* [36]. These

results indicate that *ARF* is closely related to growth and development, stress response and other physiological processes in plants.

In previous studies, *BpARF1* has been shown to be associated with drought stress. Under drought stress conditions, *BpARF1* RNA interference (RNAi)-inhibited plants presented reduced reactive oxygen species (ROS) accumulation, and enhanced peroxide (POD) and superoxide dismutase (SOD) activities. On the contrary, the overexpression of *BpARF1* showed a completely opposite phenomenon [37]. However, there are few studies on the function of abiotic stress on the ARF protein in sandalwood. Therefor, in this study, we used quantitative qRT-PCR to analyze the expression profiles of *SaARF* genes under drought. At the same time, we found that the content of auxin is extremely high in the early stage of haustorium growth and plays an irreplaceable role in the formation of haustorium [2,9]. The haustorium development in *S. album* was promoted by IAA and inhibited by the auxin biosynthesis inhibitor L-Kyn (L-kynureninean, an auxin biosynthesis inhibitor) and the polar auxin transport inhibitor NPA, indicating the haustorium development in *S. album* was enhanced by auxin synthesized in the root and/or transported from shoots [9]. To systematically research the related functions and physical and chemical properties of the *S. album ARF* gene family, we conducted an analysis of the *S. album ARF* gene family through the whole genome of *S. album* and screened 18 *SaARF* genes in this study. Detailed studies on the physicochemical properties, evolutionary relationships, gene structure, chromosomal location, collinear analysis, cis-reactive elements and tissue-specific expression of *S. album ARF*s will help to better understand the auxin response mechanism in sandalwood and provide a reference for the biological functions of *ARF* genes in response to drought stress in sandalwood.

## 2. Materials and Methods

### 2.1. Plant Materials and Treatments

*S. album* plants were grown at 25 °C in a greenhouse with a 16/8 h light-dark cycle. Four-month-old *S. album* plants were taken, and four different tissues of the plants, including mature leaves, steam, roots and haustoria collected. Twelve plants were exposed water-limited treatment, ranging from 3 to 9 d of drought. Three biological replicates were performed. Finally, they were stored at −80 °C until RNA extraction.

### 2.2. Complete Genome Identification and Sequence Analysis of the ARF Gene Family in S. album

PFAM (http://pfam.xfam.org/, accessed on 12 July 2022) was used to download the protein hidden Markov model Auxin_resp.hmm (protein family: PF06507). The hmmsearch program of HMMER3.0 software was used to compare and screen the whole genome of *S. album* proteins identified from the Research Institute of Tropical Forestry, Chinese Academy of Forestry, and protein hits with an e-value of $<10^{-5}$ and sequence score of "best 1 domain" >100 were collected [38]. We used CD Search NCBI (https://www.ncbi.nlm.nih.gov/Structure/cdd/wrpsb.cgi, accessed on 18 July 2022) to determine whether the *ARF* gene structure type and quantity of fields were screened and performed a secondary screening of the *SaARF* gene families. Protparam (https://web.expasy.org/protparam/, accessed on 12 August 2022) prediction has been used to screen protein sequences related to physiological and biochemical indicators [39], including the quantity of amino acids (aa), isoelectric point (PI), molecular weight (MW), total average hydrophobicity, instability and aliphatic index, and Wolf PSORT organelles (https://www.genscript.com/wolf-psort.html, accessed on 22 August 2022), which were used to predict the *SaARF* gene positioning. We used MEME (https://meme-suite.org/meme/tools/meme, accessed on 16 July 2022) to predict *SaARF* gene family members of the conservative base sequence and set the length between 6–50 aa and the base sequence number to 15 [40].

### 2.3. Establishment of Phylogenetic Tree

The MUSCLE method of MEGA-X was used to compare the multiple sequences of 62 *ARF* genes in *S. album*, poplar and *Arabidopsis thaliana*. The phylogenetic tree was

constructed using the Substitutions Type: Amino acid. Model: Poisson model; Rates among Sites: Uniform Rates; The Pattern among Lineages: Same (Homogeneous). After the evolutionary tree was built by iTOL, we undertook the beautification program (https://itol.embl.de/itol.cgi, accessed on 10 September 2022) [41].

### 2.4. Chromosome Mapping and Gene Structure Analysis

The chromosome length information of *S. album* and the location information of the *SaARF* gene were obtained from the whole genome annotation file of *S. album* determined by the Institute of Tropical Forestry, Chinese Academy of Forestry, and the chromosome location map was drawn with Map Draw software [42]. GSDS2.0 (http://gsds.gao-lab.org/index.php, accessed on 16 August 2022) was used to map the genetic structure of the *SaARF* gene family, including the number of CDS, introns, and UTR and their relative positions on the genes.

### 2.5. Collinearity Analysis within and between Species

In this paper, in order to more intuitively observe and analyze the evolution and genetic relationship of the *ARF* gene in different species, we not only analyzed the tandem replication and fragment replication events of the *ARF* genes in *S. album* but also compared the collinearity analysis of *ARF* among three species of *S. album*, poplar and *Arabidopsis*. The chromosome positions of *S. album* were extracted from the whole genome annotation file of *S. album*, which was provided by the Institute of Tropical Forestry, Chinese Academy of Forestry. *Arabidopsis thaliana* and poplar whole-genome files and note documents are derived from NCBI (https://www.ncbi.nlm.nih.gov/, accessed on 18 July 2022). The protein sequences of the three species were integrated, all proteins were searched by a basic local alignment search toolP (BLASTP) local search, and then tandem repeats and chromosome fragment repeats were obtained according to the results of MCScanX software [43]. Duplicate records containing the *ARF* gene were extracted by custom scripts, and finally, Dual Synteny Plotter and Advanced in TBtools (https://github.com/CJ-Chen/TBtools, accessed on 22 August 2022) were used. The Circos program was used to conduct a collinearity analysis among the three species and within the *S. album* species [44].

### 2.6. Analysis of Cis-Reactive Elements in the Promoter

The sandalwood gene structure annotation file and whole genome file were analyzed by TBtools, and 2000 bp upstream CDS of 18 *SaARF* genes were extracted. Submitting the sequence to the PlantCARE website (http://bioinformatics.psb.ugent.be/webtools/plantcare/html/, accessed on 22 October 2022), the promoter of cis reaction components was used to forecast and analyze and screen the required cis-reaction components [45]. The Simple BioSequence Viewer program of TBtools software was used to derive the distribution map of cis-acting elements in the promoter.

### 2.7. Expression Profiles of SaARF Genes in Different Plant Tissues and Drought-Treatment

An Omega kit was used to extract the total RNA from *S. album* haustoria, leaves, roots, stems and the leaves, which had been under drought conditions for 0 d, 3 d, 9 d. The quality and quantity of DNA-free total RNA was assessed using a NanoDrop ND-1000 spectrophotometer (Nanodrop Technologies, Wilmington, NC, USA). RNA samples with an A260/A280 ratio between 1.8 and 2.2 and an A260/A230 ratio greater than 2.0 were used for the subsequent analysis. According to the Takara RR036A PCR kit (Takara, San Jose, CA, USA), approximately 1–2 µL of RNA, RNase-free delH$_2$O, and 5X PrimeScript Rt Master were centrifuged and reverse-transcribed into cDNA in a T100$^{TM}$ Thermal Cycler (Bio-Rad, Hercules, CA, USA). The machine program was set at 37 °C for 15 min, 85 °C for 5 s, and stored at 4 °C. The successfully synthesized cDNA was diluted with RNase-free delH$_2$O at a ratio of 1:10 and stored at −20 °C until later use.

Finally, using cDNA as a template, real-time PCR was performed according to SYBR qPCR Master Mix (Universal) (TOLOBIO, Shanghai, China) instructions. The 18 upstream

primers and downstream primers of the *SaARF* gene were designed using Primer 3.0 (https://primer3.ut.ee/, accessed on 15 August 2022) (Table S3). The product size ranged from 100 bp to 150 bp, and the designed reaction system was as follows: 2×Q3 SYBR qPCR Master Mix 10 μL, upstream and downstream primers 0.5 μL each, cDNA template 1 μL, RNase-free delH$_2$O 8 μL. qRT-PCR was performed on a real-time PCR system based on the SYBR Green II method. Reaction procedure: Preincubation 95 °C for 900 s; Amplification 40 cycles of 95 °C for 10 s, 60 °C for 10 s and 72 °C for 20 s; Melting 95 °C for 10 s, 65 °C for 60 s and 97 °C for 1 s. Each experiment was performed with 3 biological replicates and 3 technical replicates, and the *S. album* housekeeping gene Actin was used as a reference. Relative gene expression was calculated using the $2^{-\Delta\Delta Ct}$ method [46]. Finally, SPSS Statistics 27 software was used for gene significance analysis.

## 3. Results

### 3.1. Identification and Sequence Analysis of the SaARF Gene Family

The known hidden Markov model of the *ARF* gene (Auxin_resp.hmm) was compared with the whole genome protein sequence of *S. album* determined by the Guangzhou Institute of Tropical Forestry by hmmsearch, and protein hits with an E-value of <10$^{-5}$ and sequence score of >100 were collected [38]. CD Search was used to predict the conserved domains of the selected *SaARF* candidate genes, and a total of 18 *SaARF* genes were finally obtained according to the comparison of conserved domains. According to its evolutionary relationship with the *ARF* gene in *Arabidopsis thaliana*, we named them SaARF1-SaARF18.

### 3.2. Gene Information, Conserved Domains and Conserved Motifs of the SaARF Gene Family

The CDS length of *SaARF*s is 1771 bp-3322 bp, the encoded protein generally contains 590–1111 amino acids, and the molecular weight ranges from 65.5 kDa to 122.9 kDa. The aliphatic index of SaARF was between 63.77 and 78.33, and the total average hydrophobicity of all *SaARF* genes was negative, indicating that all *SaARF* genes were encoded hydrophilic proteins. The protein instability index of SaARF was between 40.67 and 70.11, indicating that all *SaARF* genes encoded unstable proteins. Wolf PSORT was used to predict the subcellular localization of SaARF, and all SaARFs were found to be located in the nucleus (Table 1). MEME (https://meme-suite.org/meme/tools/meme, accessed on 18 July 2022) was used to analyze the protein sequence of the *SaARF* gene family, the number of motifs was determined to be 15, and the conserved motifs were analyzed (Figure 1).

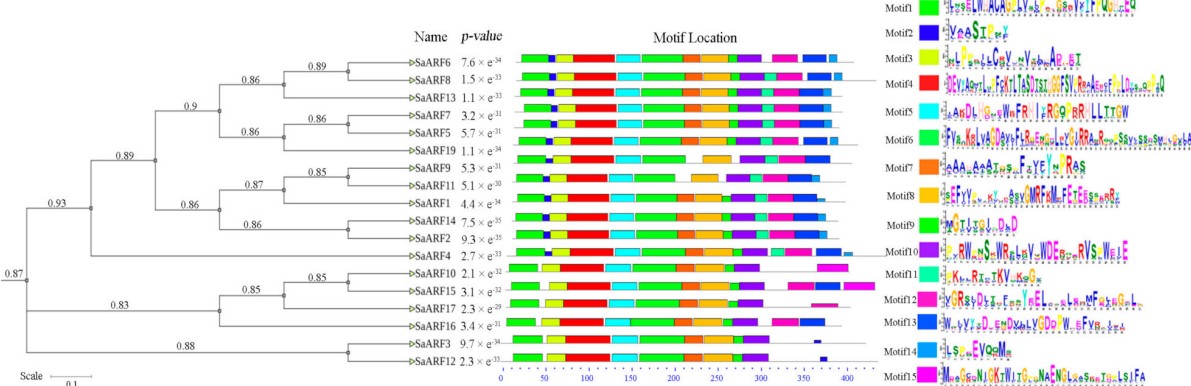

**Figure 1.** Conserved motif analysis and phylogenetic tree of gene families. Analysis diagram of conserved motifs of the *ARF* gene family in sandalwood. Phylogenetic tree and distribution map of conserved motifs of the *SaARF* gene family. The conserved motifs of the *SaARF* gene family were analyzed by MEME, and the length of the conserved motifs was set to 6–50 aa. The number was 15, and they were distinguished by different colors. Multiple sequence alignment of *S. album* ARF proteins was performed using MUCLE, and adjacency (NJ) trees were constructed using MEGA X to combine conserved motif analysis with evolutionary trees of the *SaARF* gene family.

**Table 1.** Characteristics of putative genes encoding auxin response factors in *Santalum album* L.

| Gene Name | Name | CDS Length (bp) [1] | Amino Acids (aa) [2] | Molecular Weight (kDa) [3] | PI [4] | Conserved Domains | In Silico Prediction Wolf PSORT | Instability Index | Aliphatic Index | Grand Average of Hydropathicity |
|---|---|---|---|---|---|---|---|---|---|---|
| *Sal9G26000.1* | SaARF1 | 1993 | 668 | 74.5 | 6.02 | DBD, MR, PB1 | nucl | 60.11 | 70.27 | −0.488 |
| *Sal4G08680.1* | SaARF2 | 2506 | 839 | 93.3 | 6.3 | DBD, MR, PB1 | nucl | 55.42 | 63.77 | −0.661 |
| *Sal6G09460.1* | SaARF3 | 1871 | 626 | 68.9 | 6.45 | DBD, MR | nucl | 59.42 | 75.5 | −0.34 |
| *Sal7G11880.1* | SaARF4 | 2403 | 804 | 88.9 | 5.75 | DBD, MR, PB1 | nucl | 55.05 | 75.12 | −0.419 |
| *Sal8G13630.1* | SaARF5 | 2413 | 808 | 89.1 | 5.38 | DBD, MR, PB1 | nucl | 56.65 | 73.82 | −0.375 |
| *Sal8G00600.1* | SaARF6 | 2707 | 906 | 100.4 | 6.29 | DBD, MR, PB1 | nucl | 70.11 | 70.92 | −0.493 |
| *Sal1G06860.1* | SaARF7 | 2539 | 850 | 93.9 | 5.58 | DBD, MR, PB1 | nucl | 54 | 74.44 | −0.4 |
| *Sal6G12670.1* | SaARF8 | 2557 | 856 | 94.9 | 5.74 | DBD, MR, PB1 | nucl | 66.19 | 72.91 | −0.396 |
| *Sal4G03070.1* | SaARF9 | 1873 | 628 | 71.1 | 6.33 | DBD, MR, PB1 | nucl | 50.12 | 77.07 | −0.488 |
| *Sal7G11650.1* | SaARF10 | 2123 | 708 | 77.70 | 5.91 | DBD, MR | nucl | 40.67 | 78.33 | −0.241 |
| *Sal9G20630.1* | SaARF11 | 2067 | 693 | 76.8 | 5.85 | DBD, MR, PB1 | nucl | 51.42 | 74.85 | −0.423 |
| *Sal5G03150.1* | SaARF12 | 2069 | 693 | 76.1 | 5.78 | DBD, MR | nucl | 52.15 | 69.96 | −0.397 |
| *Sal5G11390.1* | SaARF13 | 2650 | 887 | 97.90 | 6.13 | DBD, MR, PB1 | nucl | 67.49 | 75.56 | −0.398 |
| *Sal9G19410.1* | SaARF14 | 2521 | 844 | 93.6 | 6.27 | DBD, MR, PB1 | nucl | 56.52 | 66.5 | −0.611 |
| *Sal9G26810.1* | SaARF15 | 2111 | 704 | 77.2 | 7.26 | DBD, MR | nucl | 49.51 | 73.39 | −0.348 |
| *Sal2G02230.1* | SaARF16 | 1983 | 661 | 73.4 | 6.35 | DBD, MR, PB1 | nucl | 55.49 | 72.45 | −0.349 |
| *Sal6G20930.1* | SaARF17 | 1771 | 590 | 65.5 | 6.59 | DBD, MR | nucl | 47.07 | 73.02 | −0.447 |
| *Sal6G06080.1* | SaARF18 | 3322 | 1111 | 122.9 | 6.02 | DBD, MR, PB1 | nucl | 66.18 | 72.49 | −0.576 |

[1] CDS Length; [2] Length of the amino acid sequence; [3] Molecular weight of the amino acid sequence; [4] Isoelectric point of the SaARF.

Multisequence comparison of the *SaARF* gene family was performed by DNAMAN software. CD Search in the National Center for Biotechnology Information (NCBI) search domain (https://www.ncbi.nlm.nih.gov/Structure/cdd/wrpsb.cgi, accessed on 18 July 2022) has been the conservative structure prediction with its conservative structure domain. After CD Search was used to predict the conserved domains of 18 SaARFs, SaARF3, SaARF10, SaARF12, SaARF15 and SaARF17 were found to contain only two conserved domains, while the C-terminal PB1 domain was missing. The remaining 13 SaARFs all contained three conserved domains. They are DBD, MR and PB1, respectively (Figure 2A,B). Gene structure analysis, including CDS, UTR, and intron, was performed by GSDS2.0 (http://gsds.gao-lab.org/index.php, accessed on 16 August 2022) (Figure 3).

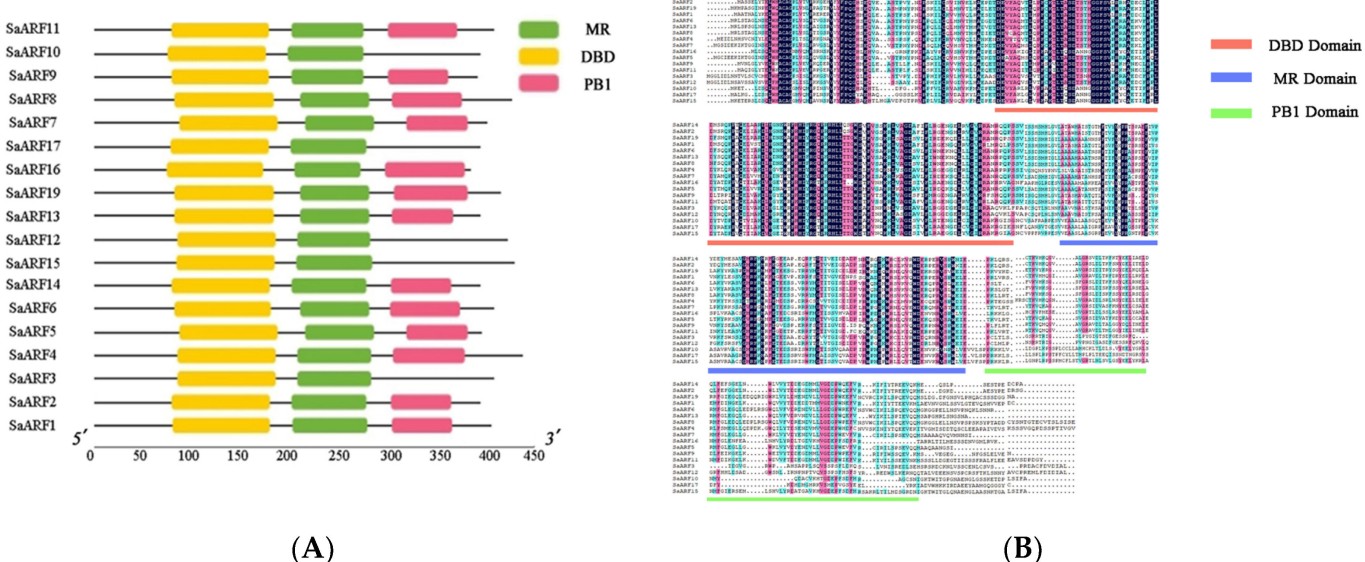

(**A**)         (**B**)

**Figure 2.** Multiple sequence comparison and conserved domain analysis of the *SaARF* gene family. (**A**) The conserved domains of the *S. album* ARFs protein were analyzed by TBtools, where the DBD domain is shown in yellow, the MR domain is shown in green, and the PB1 domain is shown in red. (**B**) DNAMAN was used to compare multiple sequences of *S. album* ARF protein and annotate the conserved domains. The DBD domain is shown in brick red, the MR domain is shown in blue purple, and the PB1 domain is shown in green.

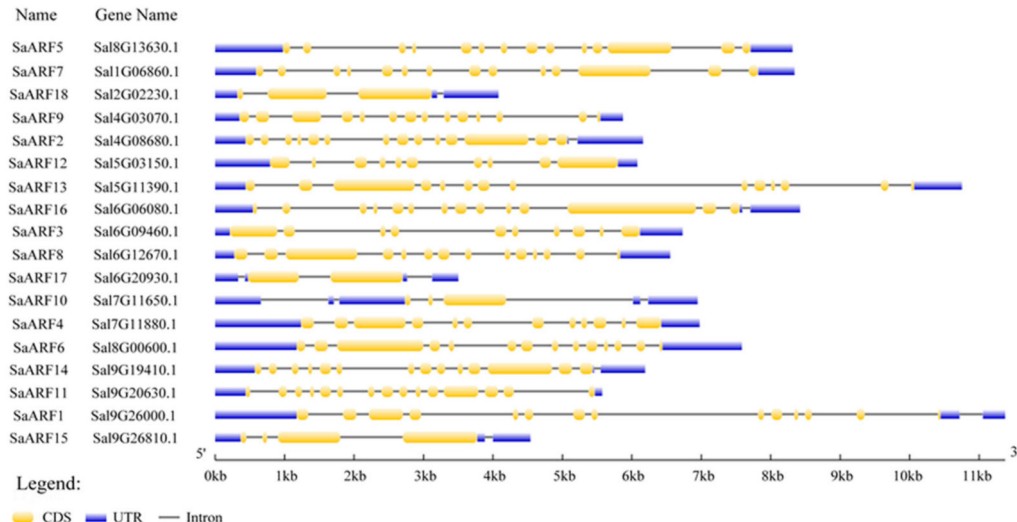

**Figure 3.** Genetic structure of the *ARF* gene in *S. album*. The value includes CDS, UTR, and intron. The yellow value is the CDS coding area, and the blue value is the UTR coding area.

### 3.3. Phylogenetic Analysis of ARF Genes in S. album, Arabidopsis thaliana and Populus trichocarpa

To better analyze the function of *S. album ARF* and its evolutionary relationship, we used the neighbor-joining (NJ), maximum likelihood and maximum parsimony method (Figures S1 and S2) to establish an evolutionary tree model of ARF proteins among *S. album*, *Arabidopsis thaliana* and *Populus trichocarpa* to observe the evolutionary relationship of *ARF*. According to the phylogenetic trees of the three species, 62 *ARF* genes were divided into four subgroups, namely, I, II, III and IV. Among them, group I included 20 *ARF* genes, including *SaARF1, 2, 9, 11, 14, 15*; there were eight *ARF* genes in group II, including *SaARF3, 4, 12*; and there were 20 *ARF* genes in group III, including *SaARF5, 6, 7, 8, 13, 18*. Group IV included 14 *ARF* genes, including *SaARF10, 16, and 17* (Figure 4).

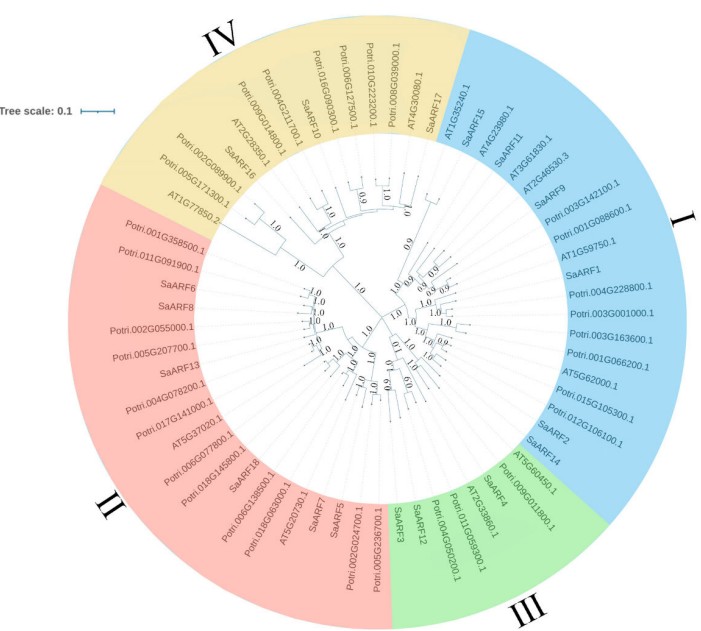

**Figure 4.** Phylogenetic relationships of *ARF* gene family members in *Arabidopsis thaliana*, *Populus trichocarpa* and sandalwood. The MEGA X program in the NJ connection method was used to construct a phylogenetic tree, which was used to perform beautification (https://itol.embl.de/itol.cgi, accessed on 16 August 2022).

### 3.4. Chromosome Mapping and Collinearity Analysis

The results of chromosome mapping showed that these genes were distributed on eight chromosomes of *S. album*, namely, chr1, 2, 4, 5, 6, 7, 8 and 9. Four *SaARF* genes were located on chr6 and chr9. Chr4, 5, 7 and 8 each contained two *SaARF* genes, chr1 and chr2 each contained an *ARF* gene, but chr3 and chr10 did not contain *ARF* genes (Figure 5). Tandem replication and fragment replication are important methods of gene expansion, and collinearity analysis can more clearly observe the results of duplication between genes. The occurrence of different members of the same gene family in the same or adjacent intergenic regions can be defined as tandem events. Through the construction of collinear maps between *S. album* and poplar and *Arabidopsis*, we found that 9 and 15 *SaARF* genes had syntenic relationships with *ARF*s of *Arabidopsis* and poplar, respectively, suggesting that some *SaARF* genes may originate from tandem or fragment replication (Table S1) (Figure 6A,B). Through the collinearity analysis among species, we found that ten of the eighteen *ARF* genes were involved in five segmental duplication events (*SaARF3/SaARF12*, *SaARF2/SaARF14*, *SaARF6/SaARF8*, *SaARF16/SaARF4*, *SaARF15/SaARF10*) in *S. album* (Table S2) (Figure 7).

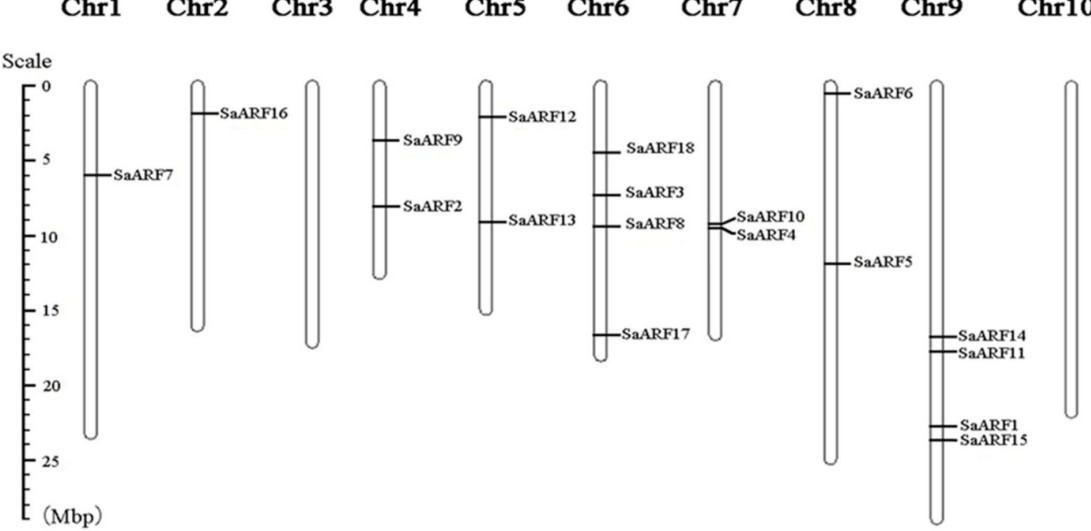

**Figure 5.** The chromosome location map of the *S. album ARF* gene was drawn using the MapDraw program, the gene name was marked on the right side of the chromosome, and the chromosome length unit was Mbp.

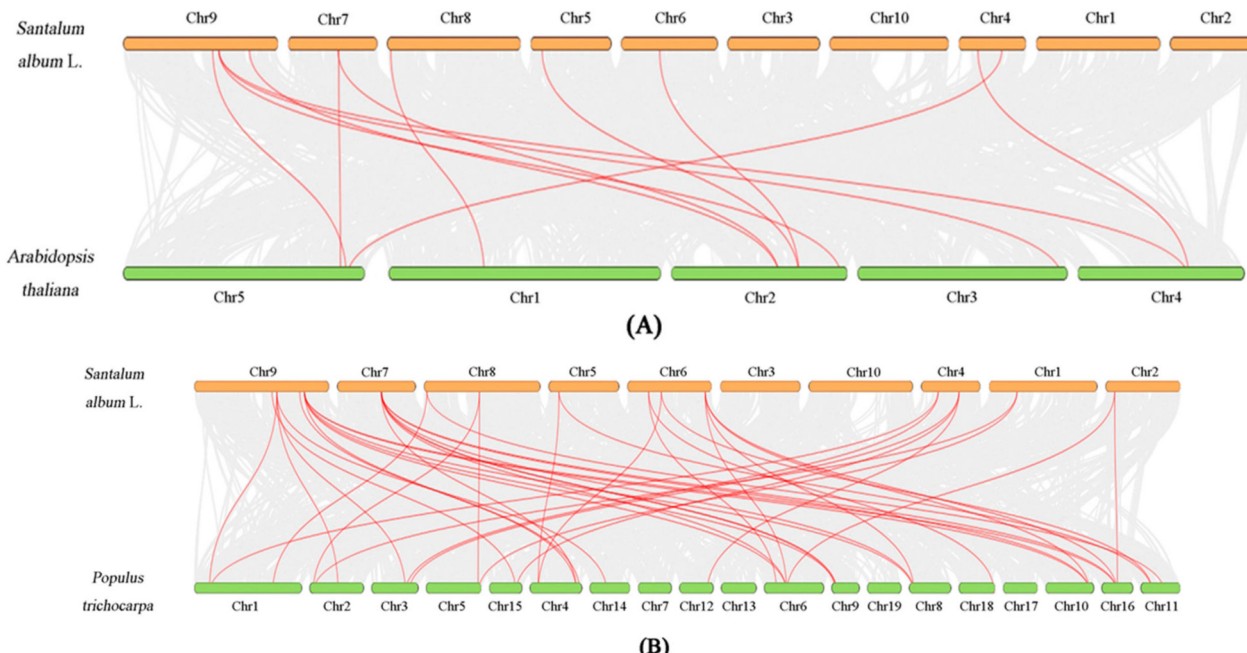

**Figure 6.** Collinearity contrast between *S. album* and two representative plants, poplar and *Arabidopsis*. (**A**) Collinearity analysis of *S. album* and *Arabidopsis thaliana*. (**B**) Collinearity analysis diagram of *S. album* and *Populus trichocarpa*. The gray lines in the background represent collinear blocks within the *S. album* and *Arabidopsis* and poplar genomes, while the red lines highlight collinear *ARF* gene pairs.

### 3.5. Analysis of Cis-Reactive Elements in the Promoter

Cis-regulatory elements in promoter sequences are essential for the temporal, spatial, and cell-specific control of gene expression, and there is evidence that genes with similar expression patterns contain the same regulatory elements in their promoters. Here, we used the 2000 bp upstream region of the 5′-UTR upstream sequence of all *S. album ARF* genes in the PlantCARE database to screen 18 cis-acting elements associated with stress, growth and development, and phytohormone responses (Figures 8 and 9). Among them, ARE cis-response elements affecting anaerobic induction were found in the promoters

of 14 *SaARF* genes, and there were auxin response elements TGA-element and AuxRR in the promoters and CAT-box in the meristem formation response elements [43]. These results indicate that the *ARF* gene has a very close relationship with plant growth and development and stress.

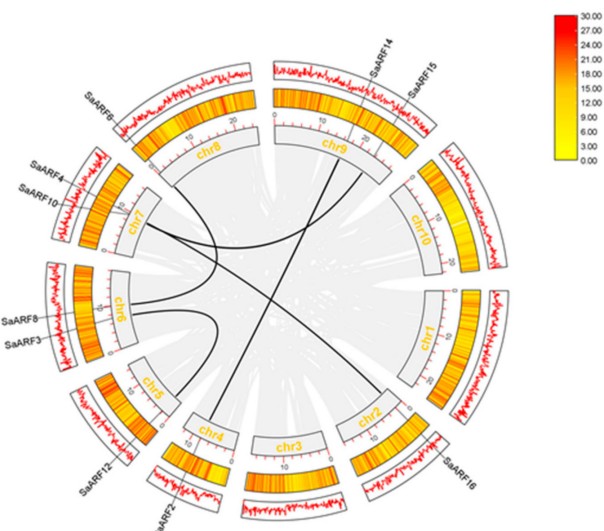

**Figure 7.** Interchromosomal relationship of the *ARF* gene in *S. album*. The gray lines indicate synteny blocks in the *S. album* genome, and the black lines indicate synteny blocks where the *ARF* repeat gene pairs are located. Chromosome names are shown in the middle of each chromosome, and the unit of chromosome length is Mbp. The golden yellow circle is the heatmap of gene density, and the outermost circle is the line map of gene density.

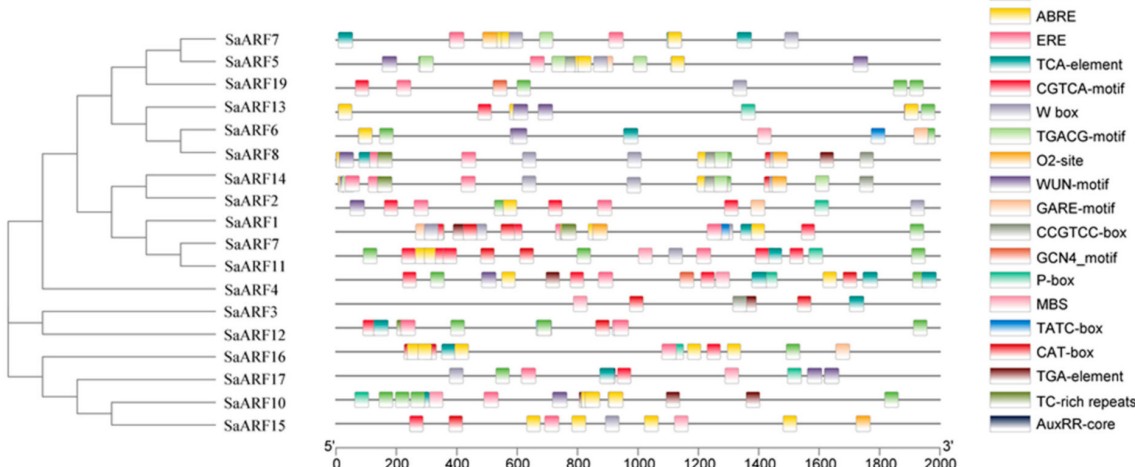

**Figure 8.** Distribution of cis-reactive elements in promoters of the *SaARF* gene family. There are 18 cis-reaction elements, including ARE, ABRE and ERE. ARE: anaerobic induction; ABRE: abscisic acid responsiveness; ERE: Estrogen Response Element; TCA-element: salicylic acid responsiveness; CGTCA-motif: MeJA-responsiveness; W-box: Trauma and pathogen reactivity; TGACG-motif: MeJA-responsiveness; O2-site: zein metabolism regulation; WUN-motif: wound-responsive element; GARE-motif: gibberellin-responsive element; CCGTCC-box: Meristem specific activation; GCN4-motif: endosperm expression; P-box: gibberellin-responsive element; MBS: MYB binding site involved in drought-inducibility; TATC-box: gibberellin-responsiveness; CAT-box: meristem expression; TGA-element: auxin-responsive element; TC-rich repeats: defense and stress responsiveness; AuxRR-core: auxin responsiveness (Table S4).

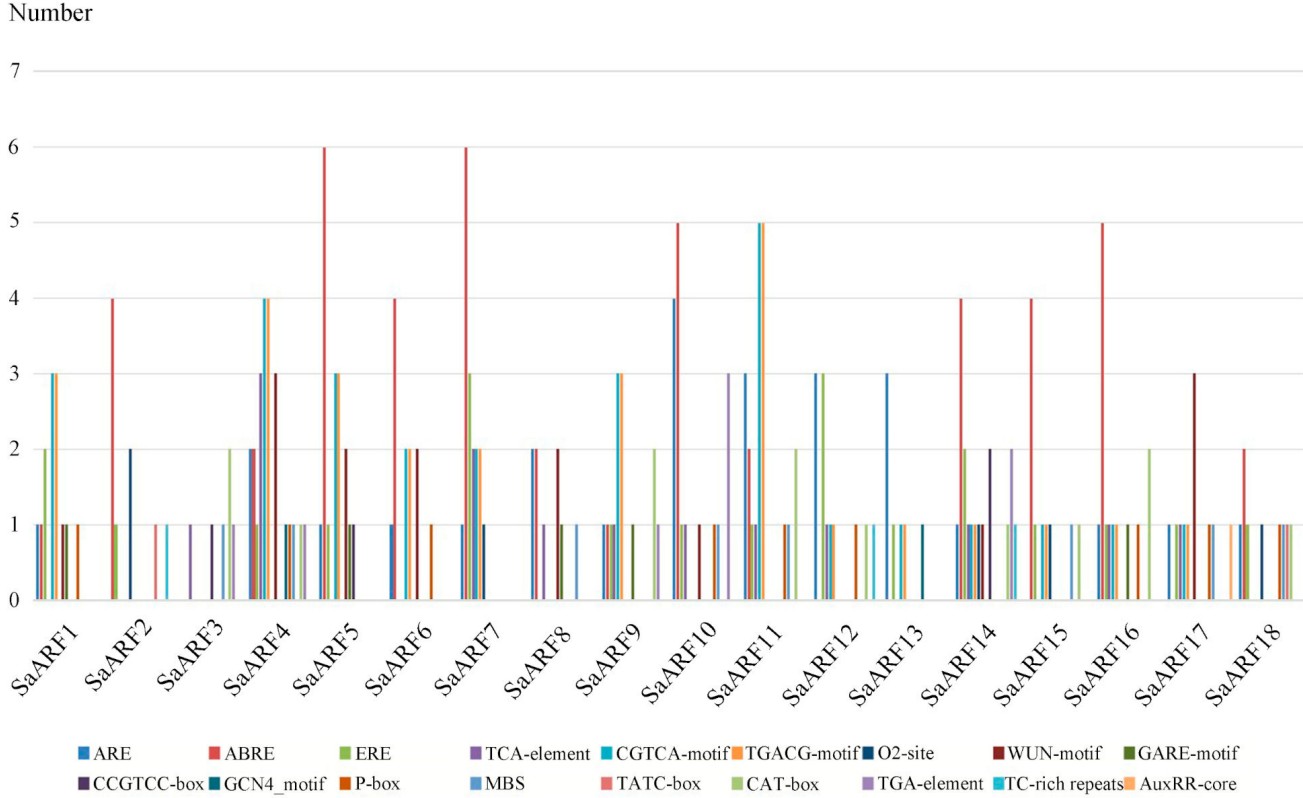

**Figure 9.** The number of cis-reactive elements of *ARF*s.

### 3.6. Tissue-Specific Expression of ARF Gene in S. album

To study the main function of *SaARF* in sandalwood plants, we used qRT-PCR to detect the tissue-specific expression of 18 sandalwood *ARF* genes and selected four main tissues, root, stem, leaf and haustoria, for analysis (Figure 10). After statistical analysis, we found that most *SaARF* genes were significantly expressed in the stem, and the expression level was high, mainly 10 genes, including *SaARF1, SaARF3, SaARF6, SaARF11, SaARF12, SaARF13, SaARF14, SaARF15, SaARF16* and *SaARF18*. The expression levels of the *SaARF1, SaARF5, SaARF7, SaARF8, SaARF15* and *SaARF16* genes were the highest in the haustorium, which was speculated to be related to the physiological process of haustorium formation or growth. However, *SaARF10* and *SaARF17* genes were not found to be expressed in the haustorium. The relative expression levels of *SaARF2, SaARF3* and *SaARF12* were the highest in leaves, while the expression levels of *SaARF9* were relatively high in roots and stems. We noticed that the expression levels of *SaARF5* and *SaARF7*, which were closely related in the evolutionary tree, were the highest in the haustorium, with significant differences. In *Arabidopsis, AtARF2* and *AtARF9* have been reported to control leaf senescence and promote lateral root elongation and leaf extension [47], respectively, and *SaARF2* and *SaARF9* genes in sandalwood were also highly expressed in leaves and roots, respectively. Therefore, it is reasonable to speculate that the *SaARF2* and *SaARF9* genes in sandalwood may have the same function, and *SaARF3* and *SaARF12*, both belonging to subgroup III, are also highly expressed in leaves. In addition, we noted that *AtARF7* has been shown to promote lateral root elongation [32], but after data statistics, we found that the expression of *SaARF7* in the haustorium was the highest, and it was reasonably predicted that *SaARF7* might be related to the growth or formation of the haustorium.

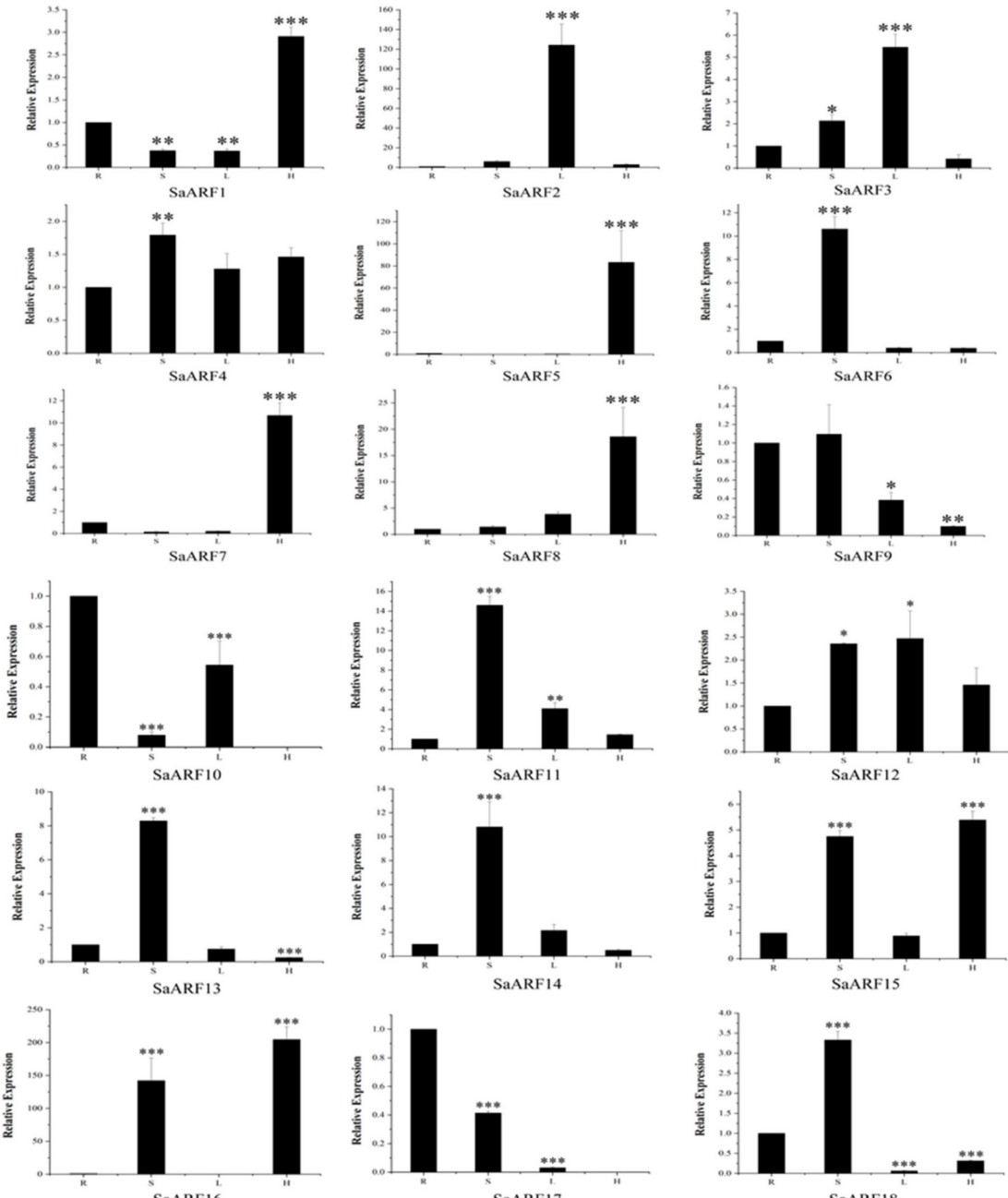

**Figure 10.** Expression analysis of 18 *ARF* genes in four representative tissues by qRT-PCR. Four various tissues include root, stem, leaf, and haustoria. The relative expression was calculated based on the $2^{-\Delta\Delta Ct}$ method. Error bars indicate ± standard deviation (SD) of three biological replicates. R: Root; S: Stem; L: Leave; H: Haustoria. Asterisks denote significant differences: * $p < 0.05$; ** $p < 0.01$; *** $p < 0.001$.

### 3.7. Expression of SaARF Genes under Drought Stress

In a previous study, *SlARF4* can affect the tomato's resistance to water shortage and *slarf4* mutants enhance plant resistance to water stress and water rehydration ability, suggesting that *SlARF4* may be an important gene in response to drought [48]. However, reports of this gene being expressed in response to drought in sandalwood are limited. Therefore, to determine whether these genes were expressed in response to drought, the expression of these 18 *SaARF* genes was also investigated during water stress. In the result, we find the expression of *SaARF2*, *SaARF5*, *SaARF10*, *SaARF16*, *SaARF17* increased significantly after 9 days (Figure 11). This suggests that it may be related to drought stress

in sandalwood. And the expression of less than half of the gene, including *SaARF10*, *SaARF11*, was upregulated and then downregulated. In the meantime, we find eleven *SaARF*s' expression level continues to decline for 3 to 9 days.

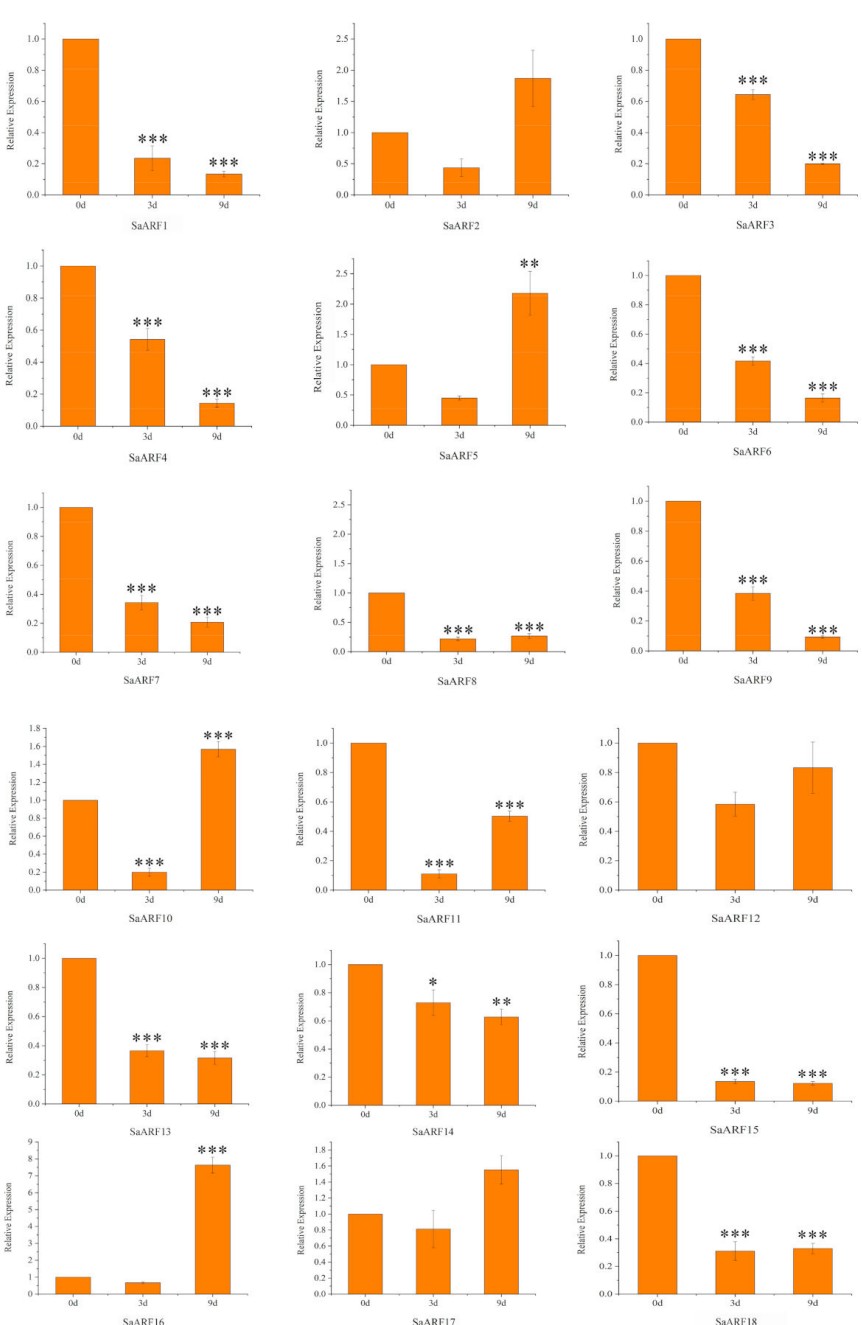

**Figure 11.** Expression analysis of Sa*ARF* genes in response to drought stress by qRT-PCR. Drought for 0 d, 3 d, 9 d in a greenhouse environment. The data were presented as the mean ± SD of three separate measurements. Asterisks denote significant differences: * $p < 0.05$; ** $p < 0.01$; *** $p < 0.001$.

## 4. Discussion

The *Arabidopsis* genome has been reported to contain 23 *ARF* genes, while the *ARF* gene families of rice, apple, poplar, tomato and eucalyptus include 25, 31, 39, 21 and 17 *ARF* genes, respectively [19,26–29]. In this study, in order to comprehend the relevant information and location of the *ARF* gene in sandalwood, we identified and characterized 18 *SaARF* transcription factors, and 18 *SaARF*s were unevenly distributed on nine chromosomes, among which chromosomes six and nine had the largest gene distribution, with four

genes on each chromosome. By using Protparam to predict and analyze the physiological and biochemical indices of the sandalwood ARF protein, we found that the subcellular localization of the SaARF protein was predicted to be located in the nucleus, and *SaARF* all encoded unstable hydrophilic proteins, indicating that it played a role in different subcellular environments.

To better understand the evolution of the *ARF* gene family of *S. album*, we characterized the conserved domains, phylogenetic relationships and collinearity of SaARF. According to the conserved domain analysis, we found that 13 of the 18 identified sandalwood ARF had three conserved domains, DBD, MR, Phox and Bem1 (PB1) [33]. The MR domain is mainly responsible for controlling the function of ARFs. The PB1 domain located at the C-terminus is homologous to the C-terminal domain of the Aux/IAA protein and can bind to form homologous or heterodimers [49]. However, SaARF3, SaARF10, SaARF12, SaARF15, and SaARF17 contain only two conserved domains and lack a C-terminal PB1 domain. All 18 SaARFs have a complete DBD domain (B3), which is consistent with previous studies. Without the B3 domain, the ARF protein has been shown to be unable to bind auxin cis-responsive elements in the promoter of auxin-responsive genes. The phylogenetic trees of 62 *ARF* genes in *Arabidopsis thaliana*, poplar and sandalwood were established, and they were divided into four subgroups: I, II, III and IV. The distribution of genes among the subclasses indicated that the expansion of the *ARF* family occurred before the divergence of the species. Each subgroup contained *ARF* genes of three species, indicating that the *ARF* genes among them had a relatively close evolutionary relationship. In addition, through the collinearity analysis within sandalwood species, we found that five segmental duplication events involved a total of 10 *SaARF* genes, which had short distance intervals and high homology in the *SaARF* gene family.

Synteny maps between two representative species and sandalwood were constructed to better understand the phylogenetic relationships. Through the collinearity analysis between sandalwood and poplar and *Arabidopsis*, we found there were 9 and 15 *SaARF* genes that had collinearity relationships with *ARF*s of *Arabidopsis* and poplar, respectively, so the evolutionary relationship between sandalwood and poplar was higher than that between *Arabidopsis* and sandalwood.

Cis-regulatory elements in promoter sequences are essential for temporal, spatial, and cell-specific control of gene expression, and there is evidence that genes with similar expression patterns contain the same regulatory elements in their promoters [50]. By analyzing the promoter cis-response elements within 2000 bp of the CDS upstream region of 18 *SaARF* genes, we found that the promoter contained anaerobically induced ARE cis-response elements, there were auxin response elements, TGA-element and AuxRR in the promoter, and the meristem formed response element CAT-box. The gibberellin response element P-box and abscisic acid response element ABRE are related to growth, development and stress [51], which indicates that the *ARF* gene is related to the realization of these functions. Among them, *SaARF3*, *SaARF4*, *SaARF10*, and *SaARF17* has both drought stress response elements and auxin response elements (Figures 9 and S5).

To investigate the main function of the *SaARF* genes and the roles of drought stress on it, we analyzed its expression in roots, stems, leaves, haustoria and the drought treatment of leaves for 0, 3, 9 days by qRT-PCR. Tissue-specific expression analysis showed that different *SaARF* genes played diverse role in the growth and development of plants. According to Figures 4 and 10, we found that *SaARF9*, *SaARF11*, *SaARF14* and *SaARF15* are highly expressed in stems which belong to group I. *SaARF5*, *SaARF7* and *SaARF8*, which are located in group II, were significantly overexpressed in the haustorium, with significant differences. At present, *AtARF7* in *Arabidopsis* has been confirmed to be related to lateral root growth, and *Arabidopsis AtARF5* has been identified as a key factor determining leaf initiation and vascular pattern formation [52]. *SaARF3* and *SaARF12* in group III and *SaARF10* and *SaARF17* in group IV were highly expressed in leaves and roots, respectively. Besides, most of the *SaARF* genes were significantly expressed in the stem, including 10 *SaARF* genes, such as *SaARF1*, *SaARF3* and *SaARF6*. After data analysis, we found that the

relative expression levels of *SaARF9*, *SaARF10* and *SaARF17* were also high in the roots, but it was notable that neither *SaARF10* nor *SaARF17* were found in the haustorium.

By comparing the expression of *SaARF* genes, in tissues were under drought stress, we found the expression of *SaARF2*, *5*, *10*, *16*, *17* were increased in 9 d, even higher than that of 0 d, especially *SaARF10*, *11*, *17* which contain drought stress response elements, but most of the *SaARF* decreased in 3/9d. This suggests that *SaARF2* are associated with a drought resistance function. Therefore, this study preliminarily revealed the expression response of *SaARF* in various tissues and under drought treatment. The response and related functions of these genes need to be further verified by experiments in further studies.

### 5. Conclusions

This is the first study on the evolutionary relationship, expression profile and putative function of the *ARF* gene in sandalwood. In the present study, we identified 18 *ARF* genes in sandalwood by domain analysis, conserved motif analysis, phylogenetic tree construction, collinearity analysis and cis-reactive element analysis and divided them into four subgroups. By analyzing the protein structure of all members of the SaARF family, we found that all *ARF* genes in sandalwood encoded unstable hydrophilic proteins, and all *SaARF* genes were distributed equally in the nucleus. Based on the qRT-PCR results, we found that most *SaARF* genes were significantly expressed in the stem (*SaARF1*, *SaARF3*, *SaARF6*, *SaARF11*, *SaARF12*, *SaARF13*, *SaARF14*, *SaARF15*, *SaARF16* and *SaARF18*). The expression levels of the *SaARF1*, *SaARF5*, *SaARF7*, *SaARF8*, *SaARF15* and *SaARF16* genes were the highest in the haustorium. Notably, *SaARF5* and *SaARF7* were specifically distributed in the haustorium, so it is reasonable to speculate that they are related to the formation or growth of the haustorium. Similarly, we found that the *SaARF10*, *16*, and *17* gene is associated with drought stress.

**Supplementary Materials:** The following supporting information can be downloaded at: https://www.mdpi.com/article/10.3390/f13111934/s1, Figure S1: A phylogenetic tree between sandalwood, *Arabidopsis thaliana* and *Populus trichocarpa* by Maximum likelihood method; Figure S2: A phylogenetic tree between sandalwood, *Arabidopsis thaliana* and *Populus trichocarpa* by Maximum parsimony method; Table S1: Collinearity analysis among species; Table S2: Collinearity analysis within species; Table S3: qRT-PCR primer sequence of 18 *SaARF*; Table S4: Cis-reactive elements in promoters of the *SaARF* gene family; Table S5: The number of cis-reactive elements in *ARF*s.

**Author Contributions:** X.L. and Y.L. designed all of the experiments and contributed to analyze the experimental results. S.M., J.L. and S.W. conceived the project. Y.C., F.Q. and D.W. helped perform the experiments. X.L., Y.L., D.W., L.H. and S.M. wrote the paper. All authors have read and agreed to the published version of the manuscript.

**Funding:** This work was supported by grants from the Fundamental Research Funds for the Central Non-profit Research Institution of CAF [CAFYBB2020SY018 and CAFYBB2019QD001], National Natural Science Foundation of China [31722012 and 31901304], Natural Science Foundation of Guangdong Province, China [2019A1515011595] and Natural Science Foundation of Chongqing, China, cstc2018jcyjAX0778.

**Data Availability Statement:** Not applicable.

**Acknowledgments:** We acknowledge everyone who contributed to this article.

**Conflicts of Interest:** The authors declare no conflict of interest.

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
