# Peer review of "Complete Genome Expression Analysis of the Auxin Response Factor Gene Family in Sandalwood and Their Potential Roles in Drought Stress"

_forests, doi:10.3390/f13111934_

Round 1
Reviewer 1 Report
It is suggested to describe the purpose or meaning physical tree building and linearity analysis using the first sentence in the Methods, such as for what? It is convenient for readers to learn, and it is also a kind of learning for authors to make clear this problem in detail.
In Methods, authors only mentioned the Poisson model when talking about the phylogenetic tree building, and did not elaborate on the specific algorithm. For closely related species and distant species, different tree building methods are used, such as maximum parsimony method, maximum likelihood method or adjacency method. The selection of models for phylogenetic tree building based on protein sequences and nucleic acid sequences is also different, which directly affects the accuracy of tree built. To solve this problem, the author needs to try to compare the results of different algorithms or models, because the determination of functional genes needs to be very careful.
The Discussion should be divided into sub topics, such as phylogenetic tree, collinearity analysis, and gene expression results etc. To a certain extent, this can also reflect the author's efforts in this study, the depth of understanding for current study, and more importantly, improve the quality of the present article.
Many English expressions in the article are too Chinese, please pay attention to improve.
Author Response
Response to Reviewer 1 Comments
Dear editor and reviewers,
We greatly appreciate both your help and that of the referees in helping me improve this paper and hope that the revised manuscript is suitable for publication. I hope this revision makes my manuscript more acceptable. We have re-submitted our manuscript and our point-by-point responses are detailed below.
Point 1: It is suggested to describe the purpose or meaning physical tree building and linearity analysis using the first sentence in the Methods, such as for what?
Response 1: Thank you for your insightful advice. The significance of establishing evolutionary tree has been mentioned in the conclusion (See line 267), and the significance of carrying out collinear analysis has been added (See line 164).
Point 2: In Methods, authors only mentioned the Poisson model when talking about the phylogenetic tree building, and did not elaborate on the specific algorithm. For closely related species and distant species, different tree building methods are used, such as maximum parsimony method, maximum likelihood method or adjacency method.
Response 2: Thank you for your suggestion. I have used the NJ method to establish the evolutionary tree in manuscript (See line 268). As for the maximum likelihood method and maximum parsimony method you proposed to establish the evolutionary tree, we have added them to the attachment(See figure S1 and S2). After analysis, we found that their results are very similar.
Point 3: The Discussion should be divided into sub topics, such as phylogenetic tree, collinearity analysis, and gene expression results etc.
Response 3: Thank you for your opinion. In response to this problem, we have adjusted the writing order of the discussion sections to make them more logical.
Point 4: Many English expressions in the article are too Chinese, please pay attention to improve.
Response 4: Thank you for your suggestion. The article has been more carefully examined and written.

Reviewer 2 Report
1)I think the figure 4 is not clear. The bootstrap values below 50% is not statistic significantly and can be omitted from the corresponding nodes.
2)It is better to show the number of cis-elements for each ARF gene, because this information is important for deriving the gene function in resistance of abiotic and biotic stresses.
3)The part of discussion should be improved.
4)The language throughout the manuscript needs further smooth.
Author Response
Response to Reviewer 2 Comments
Dear editor and reviewers,
We greatly appreciate both your help and that of the referees in helping me improve this paper and hope that the revised manuscript is suitable for publication. I hope this revision makes my manuscript more acceptable. We have re-submitted our manuscript and our point-by-point responses are detailed below.
Point 1: I think the figure 4 is not clear. The bootstrap values below 50% is not statistic significantly and can be omitted from the corresponding nodes.
Response 1: Thank you for your insightful advice. We have increased the font size of figure 4 to make it clearer and all bootstrap values above 50%.
Point 2: It is better to show the number of cis-elements for each ARF gene, because this information is important for deriving the gene function in resistance of abiotic and biotic stresses.
Response 2: Thank you for your suggestions. The number of cis-reactive element statistics have been added to the manuscript (See figure 9 and Table S5).
Point 3: The part of discussion should be improved.
Response 3: Thank you for your opinion. In response to this problem, we have adjusted the writing order of the discussion sections to make them more logical.
Point 4: The language throughout the manuscript needs further smooth.
Response 4: Thank you for your suggestion. The article has been more carefully examined and written.
